# Constitutive Correlations for Mass Transport in Fibrous Media Based on Asymptotic Homogenization

**DOI:** 10.3390/ma16052014

**Published:** 2023-02-28

**Authors:** Lukas Maier, Lars Kufferath-Sieberin, Leon Pauly, Manuel Hopp-Hirschler, Götz T. Gresser, Ulrich Nieken

**Affiliations:** 1Institute of Chemical Process Engineering, University of Stuttgart, Boeblinger Strasse 78, 70199 Stuttgart, Germany; 2German Institutes of Textile and Fiber Research Denkendorf (DITF), Körschtalstraße 26, 73770 Denkendorf, Germany; 3Institute for Textile and Fiber Technologies (ITFT), University of Stuttgart, Pfaffenwaldring 9, 70569 Stuttgart, Germany

**Keywords:** homogenization, textiles, yarns, random fibers, permeability, effective diffusion

## Abstract

Mass transport in textiles is crucial. Knowledge of effective mass transport properties of textiles can be used to improve processes and applications where textiles are used. Mass transfer in knitted and woven fabrics strongly depends on the yarn used. In particular, the permeability and effective diffusion coefficient of yarns are of interest. Correlations are often used to estimate the mass transfer properties of yarns. These correlations commonly assume an ordered distribution, but here we demonstrate that an ordered distribution leads to an overestimation of mass transfer properties. We therefore address the impact of random ordering on the effective diffusivity and permeability of yarns and show that it is important to account for the random arrangement of fibers in order to predict mass transfer. To do this, Representative Volume Elements are randomly generated to represent the structure of yarns made from continuous filaments of synthetic materials. Furthermore, parallel, randomly arranged fibers with a circular cross-section are assumed. By solving the so-called cell problems on the Representative Volume Elements, transport coefficients can be calculated for given porosities. These transport coefficients, which are based on a digital reconstruction of the yarn and asymptotic homogenization, are then used to derive an improved correlation for the effective diffusivity and permeability as a function of porosity and fiber diameter. At porosities below 0.7, the predicted transport is significantly lower under the assumption of random ordering. The approach is not limited to circular fibers and may be extended to arbitrary fiber geometries.

## 1. Introduction

Textiles are omnipresent in industrial and everyday applications, such as clothing, composite materials and construction materials. Textiles are an important class of technical materials due to their great flexibility in shape and inexpensive production processes. The performance and area of application of a particular textile often depend on its permeability. Current examples of the importance of mass transport in textiles are sportswear [1] and woven gas diffusion layers in fuel cells [2]. In sportswear, the wearing comfort is strongly influenced by air permeability and water vapor diffusion through the textile [3]; in fuels cells, the water transport in the gas diffusion layer influences the fuel cell performance [4]. Continuous filament synthetic yarns such as polyester and carbon are often used in the above applications. To adapt and optimize a given textile according to the required specifications of its application, a basic knowledge of the transport within the textile is necessary. Today, the mass transport is usually determined in experiments. However, it is not possible to distinguish between the influence of the yarn and the structure of the knitted or woven fabric on mass transfer. To improve the mass transport properties of yarns, it is desirable to study the individual influence of yarn and fabric structure separately.

Therefore, correlations based on the assumption of a regular arrangement of fibers (hexagonal and square lattice) are often used to predict the permeability of yarns [5,6,7]. However, this leads to an overestimation of the mass transfer properties [8]. In this contribution, we present new correlations for estimating the effective diffusivity and permeability of yarns as a function of porosity and fiber diameter, which take random ordering into account. To this end, we consider yarns made from continuous filaments of synthetic material. Here, the term porous or fibrous media will only refer to the yarn.

The convective and diffusive mass transport in the fibrous (porous) media can be predicted in a pore-scale simulation where the transport equations are solved on a representative cutout from μCT-images or FIB-Sem images [9]. However, pore-scale simulations are computationally very expensive and, hence are not suitable to simulate large areas of yarn material. Therefore, porous media are considered to be an effective media and transport is lumped into effective transport properties. In the context of textiles, the yarn can also be treated as an effective medium and only the weave or knit geometry is resolved in detail to simulate the mass transport in textile materials [5,10].

To model the effective convective transport in a fibrous material and in general porous media, the well-known Darcy equation is commonly applied [11]:(1)$$v=-\frac{K}{\;\eta }\cdot \nabla p,$$
where v, K, ∇p and η are the volume-averaged flow velocity, permeability tensor, pressure gradient and the dynamic viscosity of the fluid, respectively. The permeability K reflects the microstructure of the porous media, since all the convective microscale transport phenomena are mapped in the permeability K. Therefore, it is of high interest to determine the permeability K based on the structural information of the fibrous (porous) media. This relationship is valid in the creeping-flow regime (Reynolds Number ≪ 1).

One of the key structural parameters determining the permeability of fibrous materials—indeed, all porous materials—is the porosity $\phi =V_{pore}/V_{media}$, where $V_{media}$ is the total volume of the fibrous media and $V_{pore}$ is the volume not occupied by the fibers [12]. Obtained by gravimetrical measurement or imaging analysis, it is also the easiest structural parameter to identify [13]. Thus, it is of great interest to determine a constitutive permeability–porosity correlation of yarns or tows. Several researchers have published relationships of the permeability of fibrous materials as a function of their porosity for ordered fibrous media in two-dimensions or randomly oriented fiber networks in three-dimensions [8,14,15,16,17,18,19,20].

Gebart et al. [8] present an analytical, experimental and numerical investigation of the permeability of hexagonal and squared-lattice-ordered two-dimensional arrays of fibers. They derive the following correlation for $K\left(\phi \right)$:(2)$$\frac{K}{r^{2}}=C{\left(\sqrt{\frac{1-\phi_{per}}{1-\phi }}-1\right)}^{\frac{5}{2}},$$
where r is the fiber radius, $\phi_{per}$ is the critical value of porosity below which there is no permeating flow (the percolation threshold) and C is a geometric factor $C=16/\left(9\pi \sqrt{2}\right)$ and $\phi_{per}=1-\pi /4$ for a squared arrangement, and $C=16/\left(9\pi \sqrt{6}\right)$ and $\phi_{per}=1-\pi /2\sqrt{3}$ for the hexagonal-arranged fibers. The obtained correlation shows an excellent agreement to the numerical results. Since the analytical consideration assumes that the permeability is controlled by the narrow slots between the fibers, the correlation is only valid for a maximal porosity ϕ=0.65, according to Gebart et al. [8].

To study creeping flow through three-dimensional random fiber packings such as non-woven fabrics or paper-like materials, Koponen et al. [19] used the lattice Boltzmann method (LBM). Clague et al. [18] and Nabovati et al. [14] also studied the permeability of three-dimensional ordered and disordered fibrous media. They used the LBM to simulate creeping flow through fully three-dimensional random fiber networks, in which overlapping of the fibers was allowed. Based on the LBM simulations, a permeability correlation was proposed. Nabovati et al. fitted the numerical results to the constitutive permeability correlation proposed by Gebart et al. [14]. Based on asymptotic homogenization, Schulz et al. [20] studied the permeability for squared-lattice-ordered two-dimensional porous media with different geometries, such as ellipses, squares and rectangles. With these results they proposed new permeability correlations that expand the famous Kozeny–Carman equation to different geometries, since the original correlation assumes spherical particles. However, to the best knowledge of the authors, there is no constitutive permeability correlation for yarns or tows, which considers random parallel-arranged fibers.

To model the diffusive transport of a species in a fibrous media, the Ficks–Diffusion law is widely applied [21]
(3)$$J=D_{\mathrm{eff}}D\nabla \bar{c},$$
where $J,\;D_{\mathrm{eff}}$, D and $\nabla \bar{c}$ are the diffusive flux, effective diffusion coefficient, bulk diffusion coefficient and volume averaged concentration gradient, respectively. The bulk diffusion coefficient might be concentration-dependent, or dependent on the pore size in the case of Knudsen diffusion [22,23]. The effective diffusion coefficient is a structural parameter that considers the transport hinderance induced by the pore structure. For yarns and tows, it is of interest to determine a constitutive effective diffusivity–porosity correlation. Several previously published articles describe the relationships of the effective diffusion coefficient of fibrous materials as a function of their porosity for ordered fibrous media in two dimensions [24,25,26,27,28,29,30,31].

Perrins et al. [28] derived an analytical correlation for $D_{\mathrm{eff}}\left(\phi \right)$ for hexagonal and squared-lattice-ordered two-dimensional arrays of fibers, by applying the method of Lord Rayleigh [32]:(4)$$D_{\mathrm{eff}}=\frac{1}{\phi }\left(1-\frac{2\left(1-\phi \right)}{2-\phi -C_{1}{\left(1-\phi \right)}^{C_{2}}}\right)$$
where ϕ is the porosity and $C_{1}$ and $C_{2}$ are geometric factors, where $C_{1}=0.3058$ and $C_{2}=4$ for a square array, and $C_{1}=0.07542$ and $C_{2}=6$ for a hexagonal lattice. The correlation was validated numerically with different methods, such Monte Carlo simulations, asymptotic homogenization, the Voronoi tessellation method with mixing rules and by several researchers [24,25,29].

In actual yarns or tows, the fibers are not arranged in a square or hexagonal lattice, which is a basic assumption for correlations (2) and (4). In this contribution, we want to adapt the presented correlations to a randomly placed fiber setting. To do so, we apply a mathematical upscaling method.

Mathematical upscaling methods can be used to compute effective transport coefficients [33]. Helmig et al. and Battiato et al. provide a comprehensive overview of upscaling techniques [34,35]. These techniques derive macroscopic transport equations from first principles. The methods have been developed by different academic communities, such as mathematicians and engineers, and by using different methodologies. However, all methods lead to the same result for standard transport phenomena, such as diffusion and creeping single-phase flow in porous media considered here.

Asymptotic homogenization is a well-known upscaling technique based on asymptotic expansions and can be used to determine effective transport parameters for porous media. The local transport processes occurring within the porous material, as well as structural data such as porosity, influence the effective transport coefficients derived by this method. Here, we compute by asymptotic homogenization effective transport coefficients for a set of randomly generated unit cells. Curve fitting of our numerical results revealed a constitutive relationship for the permeability and the effective diffusion coefficient as a function of porosity. A similar approach has been used by Kamiński et al. and Jeulin et al. in the context of continuum mechanics and acoustics to estimate the effective material properties of composites [36,37].

The paper is structured as follows. Section 2 introduces the methodology of asymptotic homogenization and summarizes the convective and diffusive mass transport equations at the pore and continuum scale based on the homogenization theory. Additionally, the methodology of deriving a constitutive permeability–porosity correlation is presented. In Section 3, the numerical results and constitutive correlations are presented. Section 4 summarizes the results and gives an outlook on work in progress.

## 2. Methodology

The asymptotic homogenization is a mathematical averaging method that can be used to derive macroscopic transport equations, stating from a microscale description [38,39,40]. To use this method, a periodic representation on a microscopic scale must exist to represent the heterogenous media [41]. As explained in the next section in more detail, the prerequisite of periodicity can be relaxed in practical application. In this contribution, the periodic representation is denoted as Representative Volume Element Y. A cornerstone of the homogenization is the scale separation between the macroscale and the microscale. This is expressed by the size difference between the microscopic scale $l_{c}$ and the macroscopic scale $L_{c}$:(5)$$\varepsilon =\frac{l_{c}}{L_{c}}\ll 1.$$
when ε is small, the asymptotic expansion with respect to ε can be applied to the microscale description of the transport phenomena. The asymptotic expansion is
(6)$$a{\left(x\right)}^{\varepsilon }=a_{0}\left(x,y\right)+\varepsilon a_{1}\left(x,y\right)+\varepsilon^{2}a_{2}\left(x,y\right)+\cdots .$$

$a{\left(x\right)}^{\varepsilon }\;$ is spatially varying on the microscale y. $a{\left(x\right)}^{\varepsilon }$ is substituted into the equation describing the transport phenomenon on the microscale and can represent, e.g., a concentration or a velocity. The lower index here indicates the hierarchy of the scale; $a_{0}$ refers to the macroscale, $a_{1}$ to the next smaller scale, and so on. Equation (6) states that the scale becomes smaller and smaller as the index rises. In two scale asymptotic homogenization, order terms higher than $\varepsilon^{2}$ are neglected.

The limit of homogeneity is reached when ε goes to zero, at which point the heterogeneity becomes infinitely fine, as shown in Figure 1. Since there are no more structural changes in the microscopic variable and the domain is homogeneous at ε→0, the equation no longer depends on the microscopic variable y. By determining the limit ε→0, the effective transport equation of a physical process in heterogeneous media is derived by asymptotic analysis.

In the following section, we apply the asymptotic homogenization to convection and diffusion in fibrous media.

### 2.1. Representative Volume Element

In addition to a clear scale separation, the second prerequisite for the application of asymptotic homogenization is the existence of a spatially periodic domain that is representative of the porous or fibrous media. In the context of volume averaging—another upscaling method—this representative domain is also referred to as a Representative Volume Element (RVE). For more information on the concept of RVE, we refer to the classical literature [11,12,42]. As shown by several researchers, the periodicity is not a strict requirement in the sense that the structure must be strictly periodic. They have shown that a slow variation of the structural parameters over the macroscopic length still allows the application of asymptotic homogenization to derive transport parameters for porous media [22,23,43]. It has been shown also that it is possible to represent the real porous structure, and thus non-strictly spatially periodic material by artificially generated RVEs, which solely represented the characteristic structural parameters such as porosity, pore-size and width of the pore throat of the real porous media, without representing the porous structure in detail [22,43,44,45].

Based on this concept, we propose a schematic representation as shown in Figure 2. The RVE is a square domain that contains randomly periodically arranged circles. In three dimensions this can be considered as parallel fibers. By choosing a fiber diameter and fiber number, the porosity can be set. We limit the investigation to uniform nonoverlapping circular fibers, as they are most commonly found in technical textiles or composites manufactured from endless filaments from synthetic materials. Moreover, we only consider the transport perpendicular to the fibers, since for sportswear and clothing the air permeability and the water vapor transmission is usually determined perpendicular to the fibers. Our geometrical simplification is underpinned by microscopic visualization and research works of modelling water transport in yarns [13,46,47]. For yarns with twist, a three-dimensional RVE may be necessary since the transport along the fibers is influenced by the twist angle. However, we want to emphasize that the approach can be easily extended to different geometries by choosing different fiber cross-sections such as ellipses, squares or fiber-size distributions, and even a three-dimensional RVE is possible.

The RVE Y is generated using Matlab^®^, according to the workflow presented in Figure 3.

First, $2\cdot n_{fibers}$ random numbers are chosen in a range from 0 to $l_{RVE}+2r$, where $n_{fibers}$ is the number of fibers, $l_{RVE}$ is the sidelength of the RVE and r is the fiber radius. The random numbers are generated using the Mersenne–Twister algorithm and are uniformly distributed between 0 to $l_{RVE}+2r\;$ [48]. We have chosen $l_{RVE}=1$. The radius of the fibers is calculated to match the desired porosity ϕ for a given number of fibers. Each number duple is a Cartesian coordinate of the center of a fiber. If the fibers did not overlap, periodic boundaries were created, in order to apply periodic boundaries in the simulation. This was performed by checking whether the fibers violated the boundaries of the domain. If the vertical boundaries were crossed by a fiber, the fiber was mirrored by adding (for the left boundary) or subtracting (for the right boundary) $l_{RVE}$ to the x-value of the Cartesian coordinate. The same algorithm was applied to the horizontal boundaries. The Euclidean distance between the fibers was calculated subsequently. If an overlap between the fibers was observed, the procedure was started again. Due to the relatively small number of fibers in the RVE, the creation of the RVE is not very time-consuming. With this rather simple algorithm, a statistically random arrangement of the fibers in the RVE can be realized. The RVE was generated using COMSOL Multiphysics^®^’s built-in CAD functionality and meshed in COMSOL Multiphysics^®^ based on the fiber-position data generated in Matlab^®^. In Figure 3, examples of RVEs with different numbers of fibers and porosities are shown.

In the next sections, we summarize the formal asymptotic homogenization of convective and diffusive transport in porous media.

### 2.2. Homogenization of Convective Transport

The convective mass transport on the pore-scale is covered by the Navier–Stokes equations. Due to the small pores of the fibrous media and the low velocity, creeping flow (Reynolds Number ≪ 1) is a reasonable assumption, i.e., inertia can be neglected.

**Figure 4 materials-16-02014-f004:**
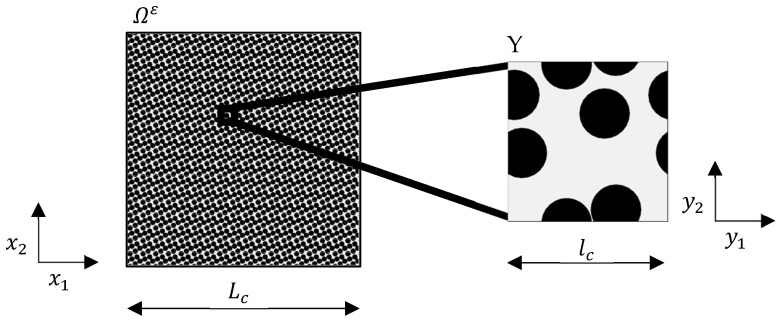
Fibrous media that is composed of spatially periodic RVE. The RVE consists of randomly arranged circles representing the fibers in two dimensions.

For a fibrous (porous) medium with impermeable walls (see Figure 4), this gives the Stokes problem the following form:(7)$$\left\{\begin{array}{c} \begin{array}{ccc} \varepsilon^{2}\eta \Delta_{y}v^{\varepsilon }- & \nabla_{y}p^{\varepsilon }=0, & in\;Y_{F}\\ & \nabla_{y}v^{\varepsilon }=0, & in\;Y_{F}\\ & v^{\varepsilon }=0, & on\;\Gamma . \end{array} \end{array}\right.$$
where $v^{\varepsilon }$, $p^{\varepsilon }$, $\eta ,\;Y_{F}$, ε are the velocity, pressure, viscosity, pore space and the order parameter of scale separation, respectively. Slip on the wall Γ of the fibers and additional body forces are neglected. In order to perform an asymptotic expansion, the viscosity is scaled by $\varepsilon^{2}$. Without scaling, frictional forces would dominate for ε→0. This means the pressure gradient would have no effect on the velocity profile. Physically, this is reflected by standstill [49]. The scaling postulates that the shear forces are in equilibrium with the frictional forces. This results in a physically reasonable solution to the problem for ε→0.

In the next step, an asymptotic expansion for the quantities $v^{\varepsilon }$ and $p^{\varepsilon }$ is performed. For this, the ansatz (6) and (7) are applied to the pore-scale transport Equation (8). This leads to the following cell problems in Y for convective mass transfer:(8)$$\left\{\begin{array}{ccc} \begin{array}{ccc} \Delta_{y}\vec{\chi_{j}}-\nabla_{y}\Pi_{j} & +\vec{e_{j}}=0, & y\in Y_{F}\\ & \nabla_{y}\vec{\chi_{j}}=0, & y\in \Gamma \\ & \vec{\chi_{j}}=0, & y\in \Gamma \end{array}\\ \;\Pi_{j}\;and\;\vec{\chi_{j}}\;are\;Y-periodic, & & \end{array}\right.$$
where the base functions $\vec{\chi_{j}}$ and $\Pi_{j}$ are the local variation of the velocity and pressure, the lower index j denotes the spatial directions, e.g., in two-dimensions $j=\left[1,2\right]$. Thus, two cell problems are solved to determine the permeability tensor in two dimensions. In addition, we assume that the porous media $\Omega^{\varepsilon }$ is composed of a spatially periodic RVE Y. The detailed derivation of the cell problem (9) can be found in [49].

The result of the two-scale asymptotic expansion is the Darcy equation:(9)$$v=-\frac{K}{\;\eta }\cdot \nabla p,$$
where v, K, ∇p and η are the volume-averaged flow velocity, permeability tensor, pressure gradient and the dynamic viscosity of the fluid, respectively. The permeability can be calculated by solving the cell problems (8) on Y (see Figure 3). By volumetric averaging of the base functions $\vec{\chi_{j}}$:(10)$$k_{ij}=\frac{1}{\left|Y\right|}\int_{Y_{F}}\chi_{ij}dy,$$
where $Y_{F}$ is the void space, the permeability tensor in, e.g., two dimensions is given by:(11)$$K=\left(\begin{array}{cc} k_{11} & k_{12}\\ k_{21} & k_{22} \end{array}\right).$$

To transform the permeability K in a dimensionless form, K is scaled by the square of the characteristic microscopic length $K/l_{c}^{2}$. We choose as the microscopic characteristic length $l_{c}=r$, where *r* is the fiber radius. To determine the permeability tensor K, the cell problem (8) in the RVE Y must be solved first. 

**Figure 5 materials-16-02014-f005:**
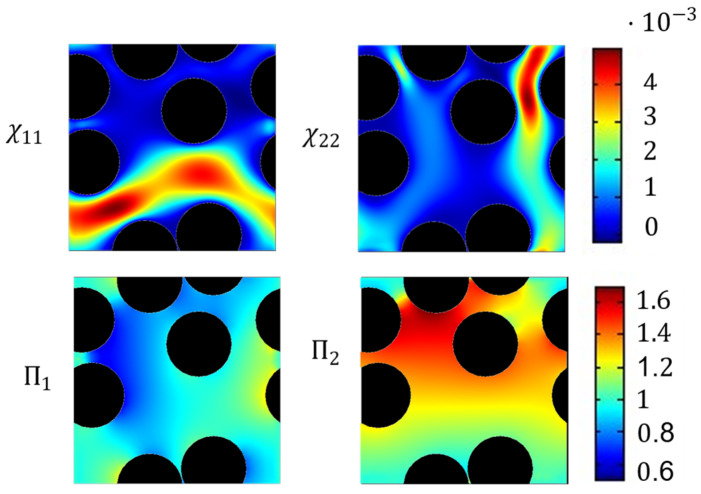
Base functions of the cell problem for convection in the RVE geometry with five fibers. On the left side, the corresponding base functions to the $y_{1}$-direction, and on the right side to the $y_{2}$-direction. **Top**: Base functions velocity $\chi_{ij}$. **Bottom**: Base functions pressure $\Pi_{j}$.

An example of the base functions is shown in Figure 5. The simulation of the cell problem was performed with the finite-element simulation software COMSOL Multiphysics^®^. On the boundaries of the domain, we applied periodic conditions. For discretization, a triangular mesh was applied, where the $\vec{\chi_{j}}$ variable was discretized with second order elements, while $\Pi_{j}$ was discretized with first order elements. The direct solver PARADISO was used to solve the cell problem [50]. The permeability tensor was calculated by surface integration. Numerical accuracy was ensured by a mesh study not shown here.

### 2.3. Homogenization of Diffusive Transport

In this section, we outline the homogenization of diffusive transport. Diffusion in the pores $\Omega_{F}^{\varepsilon }$ of the periodic porous media $\Omega^{\varepsilon }$, illustrated in Figure 4, can be modelled by the pore-scale transport equation:(12)$$-\nabla \cdot \left(D\nabla c\left(x\right)\;\right)=0\;in\Omega_{F}^{\varepsilon }.$$

D is the molecular diffusion coefficient and $c\left(x\right)$ is the concentration that is depended on the macroscopic spatial variable x. To use the asymptotic homogenization to derive an effective diffusion equation, we scale the variables by characteristic quantities: $D^{*}=D/D_{c}$, $c^{*}=c/c_{c}$, $y^{*}=y/l_{c}$ and $x^{*}=x/L_{c}$. The parameters with subscript c are the characteristic parameters, respectively. Here, we choose the molecular diffusion coefficient D as characteristic diffusion coefficient $D_{c}$. Dropping the asterisks, the scaling results in the dimensionless equations are as follows:(13)$$\left\{\begin{array}{cc} \;\;-\nabla \cdot \left(D^{\varepsilon }\nabla c^{\varepsilon }\right)=0, & in\;\Omega_{F}^{\varepsilon }\\ \;\;-\vec{n}\cdot \left(D^{\varepsilon }\nabla c^{\varepsilon }\right)=0, & on\;\Gamma^{\varepsilon }\\ \;\;\;\;\;\;c^{\varepsilon }=c_{D}, & on\;\partial \Omega_{}^{\varepsilon }, \end{array}\right.\;$$
where $\vec{n}$ is the normal vector on the pore wall $\;\Gamma^{\varepsilon }\;$ and $D^{\varepsilon }$ is the dimensionless diffusion tensor. The index ε indicates the dependence on the microscopic variable y.

By applying the expansion (6) to the pore-scale transport Equation (13), the base functions $w_{j}\left(y\right)$ are the self-similar local changes in concentration and the so-called cell problem. For a more detailed derivation, we refer to [49,51,52,53].
(14)$$\left\{\begin{array}{cc} \;\;-\Delta_{y}w_{j}=0, & y\in Y_{F}\\ \vec{n}\cdot \left(\nabla_{y}w_{j}\right)=-\vec{n}\cdot \vec{e_{j}}, & y\in \Gamma \\ \;\;\;w_{j}is\;Y-periodic. & \end{array}\right.\;$$

The result of the homogenization after redimensioning is an effective diffusion equation
(15)$$-\nabla \cdot \left(D_{eff}D\nabla \bar{c}\;\right)=0\;in\;\Omega$$

$D_{\mathrm{eff}}$ is the effective dimensionless diffusion tensor, with
(16)$$d_{ki}=\frac{1}{\left|Y\right|}\int_{Y_{F}}\left(\delta_{ki}+\frac{\partial }{\partial y_{k}}w_{j}\left(y\right)\right)dy,\;$$
where $w_{j}\;$ is the solution to the cell-problem (14). Since $w_{j}$ is integrated via $Y_{F}$, the integral refers to the averaged concentration in the void space $c_{i}$. To relate the diffusion coefficients to the volume-averaged concentration, the coefficient must be scaled by $\frac{1}{\phi }$, since $c_{i}=\phi \cdot {\bar{c}}_{i}$. The effective diffusion tensor then reads:(17)$$D_{eff}=\frac{1}{\phi }\cdot \left(\begin{array}{cc} d_{11} & d_{12}\\ d_{21} & d_{22} \end{array}\right)$$
with the averaged concentration ${\bar{c}}_{i}$ of the species i in the porous media Ω. 

**Figure 6 materials-16-02014-f006:**
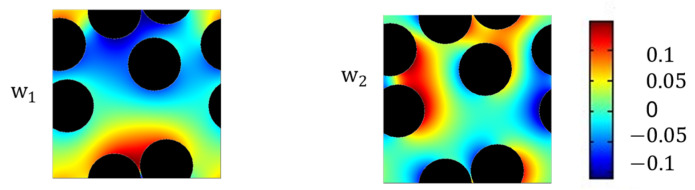
Base functions of diffusion in the RVE geometry with five fibers. $w_{1}$ is the base function of the $y_{1}$-direction and $w_{2}$ to the $y_{2}$-direction.

The computed basis functions $w_{j}$ are shown in Figure 6. As for the cell problem of convective transport, we implemented the cell problem (15) in the commercial simulation program COMSOL Multiphysics^®^ to solve the set of equations, and the same solver and boundary conditions were applied. For discretization, a triangular mesh was applied, where the $w_{j}$ base functions were discretized with second order elements. By numerical surface integration, the diffusion tensor was derived.

### 2.4. Deriving Constitutive Transport Relationships Based on Asymptotic Homogenization

In the following, we outline the approach to derive constitutive transport relationships. A schematic flow chart of the methodology is shown in Figure 7.

In the first step, the virtual RVEs are created in order to represent the characteristic microstructure of the considered material. For a fixed porosity, a representative number of RVEs are created to represent the average geometrical configuration. In the next step, the cell problems (8) and (15) are solved on a set of RVEs, which are randomly generated taking porosity and fiber diameter as a constraint. Finally, the transport properties obtained by the simulations are fitted to correlations (2) and (4). This averages the transport coefficients over all random geometries at a given porosity ϕ.

## 3. Results and Discussion

In this section, we present the simulation results and improved constitutive correlations for mass transport in fibrous media. To demonstrate the impact of random ordering, we compare our proposed correlations to correlations by Gebart [8].

### 3.1. Number of Fibers in the Representative Volume Element

To determine a representative and generally valid correlation, it is of great importance that the numerical results must be invariant to the number of fibers in the RVE. To verify this, 300-RVEs were generated for different porosities ϕ=0.45 to ϕ=0.95 with different number of fibers ($n_{fiber}=\left[5,\;6,\;7,\;8\right])$, giving in total 1500-RVE. Next, the data were averaged for the respective porosity at a fiber number. As pointed out by King et al. [54], transport coefficients for random porous media have been geometrically averaged. As can be seen in Figure 8, even five fibers are sufficient to obtain numerical results, which are independent of the number of fibers and therefore the size of the RVE.

In all calculations, we set the number of fibers in the RVE to be $n_{fiber}=5$ to minimize the computational cost.

### 3.2. Constitutive Permeability–Porosity Relationship for Random Arranged Parallel Fibers

We determine the permeability of 300 random RVEs in the x- and y-direction for each given porosity. In all RVEs, the fibers have the same radius. The porosity of the RVE varied from 0.35 to 0.99. Since the geometry is isotropic on average, 600 data points were obtained for each porosity; hence, we received 38,400 data points in total. The results are plotted in Figure 9. As expected, the permeability tends to infinity in the limit ϕ → 1 and drops towards zero at low porosity. The variation in the data is greatest at low porosities, where the random arrangement of the fibers leads to large relative changes in the predicted permeability. A modified version of the Gebart [8] relationship provided an excellent fit over the range of porosities considered. We adapt Gebart’s original relationship by adapting three constants:(18)$$\frac{K}{r^{2}}=C_{1}{\left(\sqrt{\frac{1-\phi_{per}}{1-\phi }}-1\right)}^{C_{2}},$$
where $\phi_{per}$ is the value of porosity above which flow can occur, in fact the percolation threshold. $C_{1}$ and $C_{2}$ relate to the RVE geometry. A similar approach was chosen by Nabovati et al. [14]. The fitted correlation is shown in Figure 9.

We used the lsqcurvefit function of Matlab^®^, with a Levenberg–Marquardt algorithm for minimizing the Euclidean norm. The fit is performed in logarithmic space to avoid biasing the fit towards large permeability values at high porosity. The parameters are given in Table 1.

The fitted $\phi_{per}$ is comparable in magnitude to the analytically determined percolation threshold of the squared arrangement ($\phi_{per}=0.21)$ by Gebart [8], but due to the random arrangement in our geometrical consideration, the percolation threshold reached a higher porosity, as expected. However, we emphasize our correlation is only valid in a porosity range from 0.35 to 0.99. Further validation is required for porosities below 0.35. Nevertheless, the covered porosity range is the relevant region for practical applications, since porosities close to the percolation threshold rarely occur in textiles [55].

To verify the fitting and to ensure that the optimization algorithm did not reach a local minimum when fitting the parameters, the fitted correlation was compared to the geometrical mean of the simulation data. The comparison is shown in Figure 10.

The fitted correlation almost perfectly crosses the averaged data points, and $R^{2}=0.998$ is obtained.

Direct comparison of the modified correlation with that by Gebart [8] for hexagonal and squared lattices shows the impact of the random arrangement of fibers for the predicted permeability. Figure 11 shows the comparison between the random, squared and hexagonal arrangement of the fibers. Due to the logarithmic scaling of the y-axes, the differences are quite large even though the curves are relatively close to each other.

The correlations of the ordered-arranged fibers predict lower permeability than the correlation for randomly ordered fibers below 0.8. This might explain why Gebart overestimated the permeability when compared to experimental findings for a squared arrangement.

### 3.3. Constitutive Diffusivity–Porosity Relationship for Randomly Arranged Parallel Fibers

Similar to permeability, we determined the effective diffusion coefficient of 300 random RVEs in the x- and y-direction for each porosity. The porosity of the RVE was varied from 0.35 to 0.99. Again, 38,400 data points were calculated. The simulation results are plotted in Figure 12. As expected, the effective diffusion coefficient tends towards 1 in the limit *ϕ* → 1 and drops towards zero at low porosity. As already observed for the permeability, the variation in the data is greatest at low porosities, where the random arrangement of the fibers leads to large relative changes in the predicted permeability. We found that a modified version of the Perrins [28] relationship provided an excellent fit to the data across the entire porosity range. We adapt Perrin’s original relationship by adapting two parameters.
(19)$$D_{eff}=\frac{1}{\phi }\left(1-\frac{2\left(1-\phi \right)}{2-\phi -C_{1}{\left(1-\phi \right)}^{C_{2}}}\right),$$
where $C_{1}$ and $C_{2}$ are related to the geometry of the RVE.

Again, we used the lsqcurvefit function of Matlab^®^, with a Levenberg–Marquardt algorithm for minimizing the Euclidean norm. The fitted parameters are listed in Table 2.

To validate the fit, we compared the correlation to the geometrical mean of the simulation data in Figure 13. As it was observed for the permeability, the fitted correlation runs almost perfectly through the averaged data, giving $R^{2}=0.999$.

Figure 14 shows a comparison of our proposed correlation to the correlations by Perrins [28] for fibers in a squared and a hexagonal arrangement.

The difference between the correlations is obvious. Especially for a porosity below 0.8, the difference becomes significant. Effective diffusion coefficients calculated from ordered arrangements of fibers are much larger compared to random arrangements.

## 4. Conclusions

In this work, we present improved constitutive transport correlations for diffusive and convective mass transport in yarns. The proposed correlations were determined by a new approach using digital reconstruction of the yarn and asymptotic homogenization to work out the transport parameters. We propose to generate a large number of RVEs to statistically represent the microstructure of the porous material under consideration. The cell problems arising from the asymptotic expansion are then solved on the RVEs. The constitutive transport correlations are obtained by curve fitting to calculate the transport parameters. The proposed method for deriving constitutive transport correlations can be applied in future studies to other manufactured porous materials for industrial applications, e.g., battery electrodes, catalysts and filters. The prerequisites are a clear scale separation and a porous structure that can be represented in an RVE and modified by varying the characteristic structural properties, e.g., particle-size distribution or porosity.

The newly derived correlations for yarns facilitate a more accurate prediction of the transport across the fiber banks in the yarns and tows and give a more detailed description of the transport phenomena within the structures. We adapted the correlations proposed by Gebart and Perrins to a more realistic geometric representation of real textile structures [8,28]. We compared the proposed correlations with those in the literature, where fibers are arranged in a squared or hexagonal pattern. The comparison showed that the random arrangement significantly affects transport across the fibrous media, which is in agreement with experimental results in resin transfer moulding [8]. However, the proposed correlations are only applicable to yarns with parallel oriented fibers with a circular cross-section and in a porosity range from 0.35 to 0.99. Additionally, the permeability correlation is only applicable when the flow in the yarn is in the Stokes regime (Reynolds Number ≪ 1).

In the future, we will compare the proposed correlations with experimental fabric transport measurements to investigate the influence of the yarn structure on the transport properties of the fabric.

## Figures and Tables

**Figure 1 materials-16-02014-f001:**
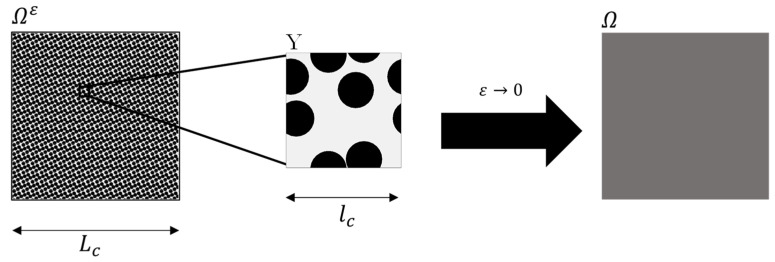
Conceptualization of the asymptotic homogenization.

**Figure 2 materials-16-02014-f002:**
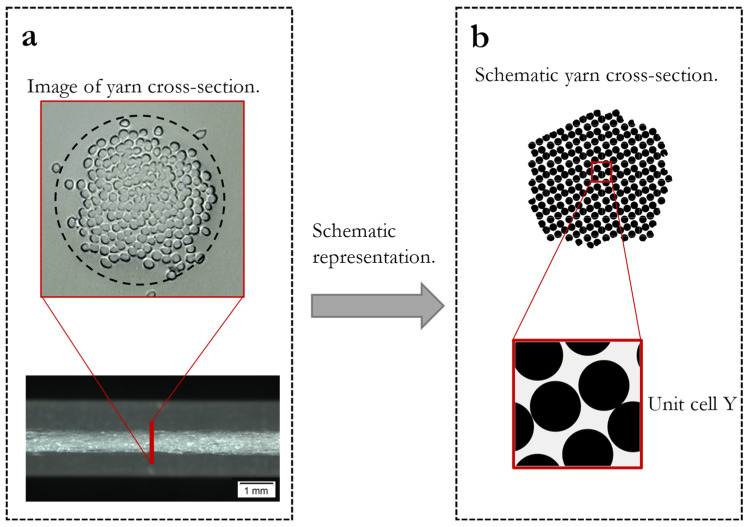
(**a**) Image of a melt spun multifiber polyester yarn and visualization of a cross-section of yarn by embedding in resin. (**b**) Schematic representation of yarn cross-section with corresponding RVE Y.

**Figure 3 materials-16-02014-f003:**
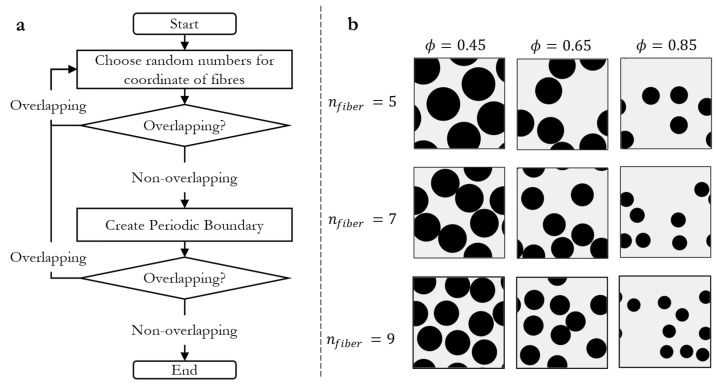
(**a**) Schematic process diagram of the generation of RVEs. (**b**) Exemplary RVEs with different porosities and fiber numbers.

**Figure 7 materials-16-02014-f007:**
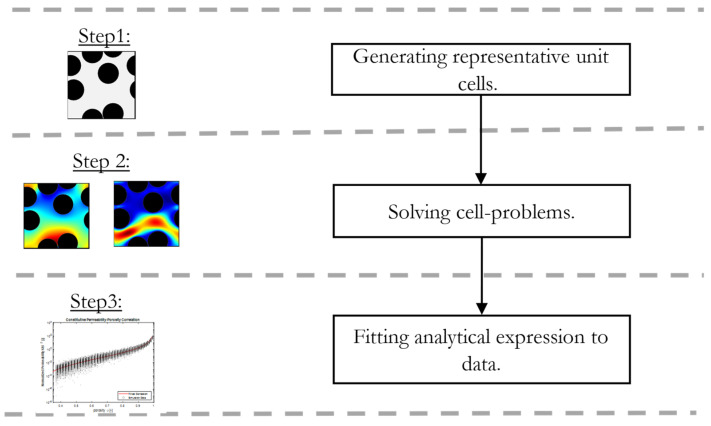
Schematic flow chart to determine constitutive correlations for mass transport in fibrous media.

**Figure 8 materials-16-02014-f008:**
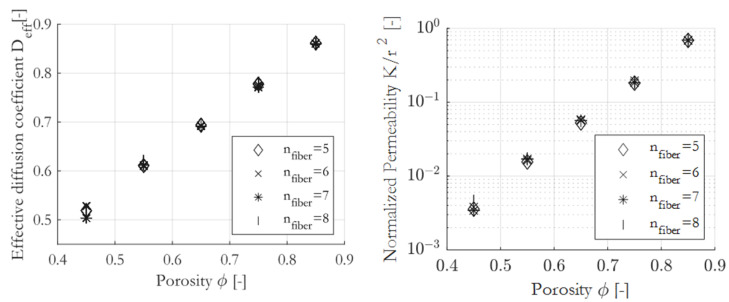
Geometrical mean of transport coefficients, computed for RVEs with a different number of fibers.

**Figure 9 materials-16-02014-f009:**
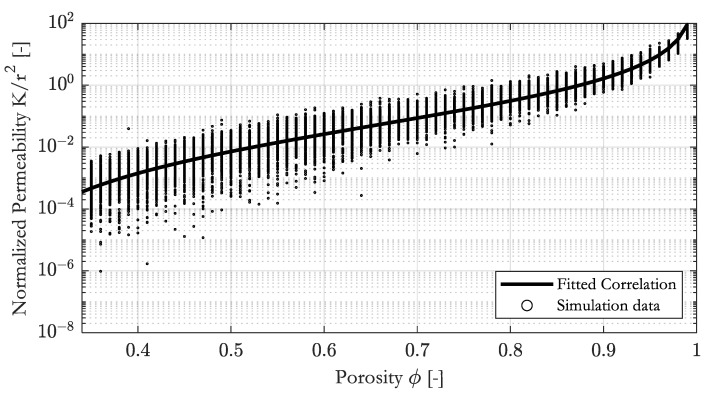
Constitutive permeability–porosity correlation and simulation data.

**Figure 10 materials-16-02014-f010:**
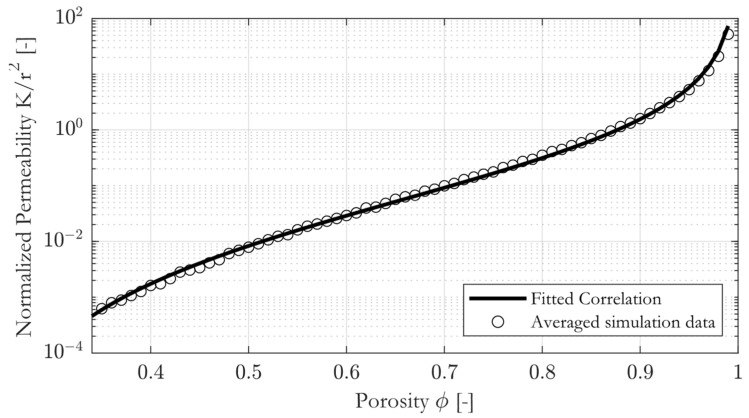
Constitutive permeability–porosity correlation with geometrical mean of simulation data.

**Figure 11 materials-16-02014-f011:**
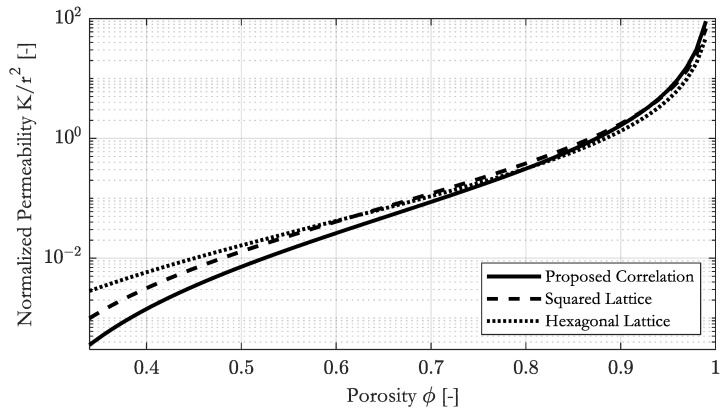
Comparison of the proposed correlations to correlations by Gebart [8] for a squared and a hexagonal arrangement of the fibers, to estimate the permeability of fibrous media.

**Figure 12 materials-16-02014-f012:**
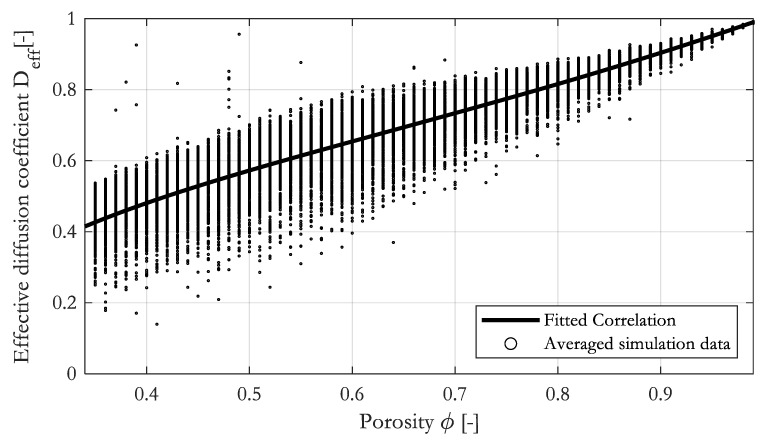
Constitutive diffusion–porosity correlation and simulation data.

**Figure 13 materials-16-02014-f013:**
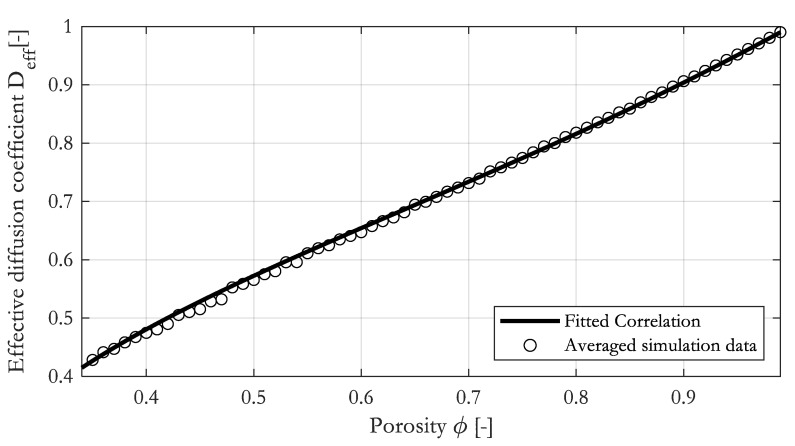
Constitutive diffusion–porosity correlation with geometrical mean of simulation data.

**Figure 14 materials-16-02014-f014:**
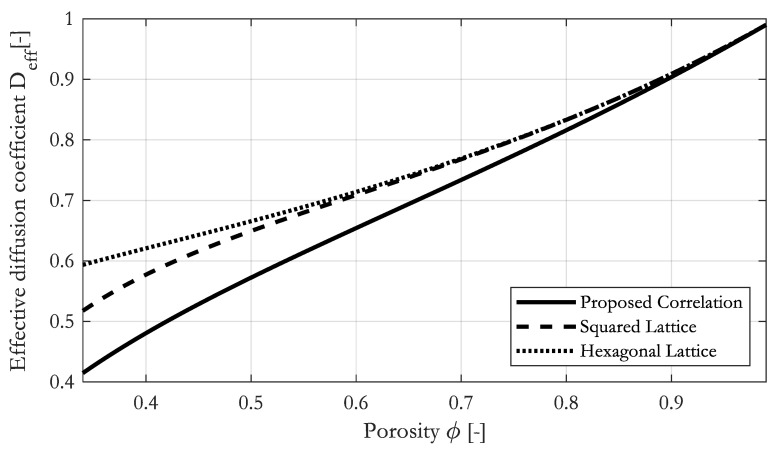
Comparison of the proposed correlation to correlations by Perrins et al. [28] for fibrous media to estimate the effective diffusion coefficient.

**Table 1 materials-16-02014-t001:** Fit values of the permeability–porosity correlation.

Parameter	Bestfit Value
$C_{1}$	0.3468
$C_{2}$	2.6193
$\phi_{per}$	0.2306

**Table 2 materials-16-02014-t002:** Fit values of the diffusion–porosity correlation.

Parameter	Best Fit Value
$C_{1}$	0.1711
$C_{2}$	0.7895

## Data Availability

The data and results involved in this study have been presented in detail in the paper.

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
