# Peer review of "Constitutive Correlations for Mass Transport in Fibrous Media Based on Asymptotic Homogenization"

_materials, 2023, doi:10.3390/ma16052014_

Round 1

Reviewer 1 Report

This manuscript reported correlations to estimate the effective diffusivity and permeability of yarns as a function of porosity for random ordered fibers. The proposed correlations are determined using a digital reconstruction of the yarn and asymptotic homogenization. However, this paper contains many weak points, which needs to be revised in light of the recommendations mentioned below.

Some major concerns are shown as following:

1. Please unify the format of the references. Some references the authors cited are missing in the manuscript.

2. Introduction: What is the novelty of your work? The authors should include the novelty of the manuscript in this part.

3. Methodology: The authors claimed that the asymptotic homogenization method was a suitable method for the determination of effective transport coefficients in porous media. However, there are many other methods for this determination. It is recommended that the authors should make a comprehensive comparison for these methods by listing a table from some specific aspects.

4. Figure 1: Why did the authors select polyester yarn for visualization? What is the spinning method for this polyester yarn? There are many spinning methods for preparation a polyester yarn. The cross-sections for these yarns derived from various spinning methods are significantly different. The authors should explain and expound more about this comment in the revised manuscript. Additionally, is this a single-subject experimental design or a statistical experiment design?

5. Equations: the authors should deduce again for the equations appeared in the manuscript.

6. Conclusions: From the results conclude by the authors, there are also some limitations for this method. The authors should address the limitations of this method in the conclusions.

Reviewer 2 Report

Dear Editor/Author;

-          Abstract should be detailed.

-          Emphasis should be placed on the purpose of the article.

-          The article should be reviewed by a native English reader.

-          Grammatical errors should be corrected throughout the article.

-          The conclusion should be strengthened.

-          The contribution of this article to future studies should be stated.

-          It should be clearly stated in which area this article fills the gap.

Reviewer 3 Report

The paper has been devoted to the asymptotic homogenization of some mass transport problems in random fibrous composites. This work is generally quite well written and could be published provided that some minor comments would be considered. These are: 

1. the Introduction needs to be completed with the works related to random composites (particularly fibrous structures) composed by Ostoja-Starzewski, KamiÅ„ski, Sakata, and Jeulin, for instance; 

2. the Authors write (cit.): "In this work, we compute for a set of randomly generated unit-cells effective transport coefficients by asymptotic homogenization." This statement needs to be completed with a detailed description of the random generation procedure. Random numbers mentioned a little bit further need to have some distribution, whose parameters need to be justified with the RVE size (by the way, literature in this area uses RVE or SVE instead of REV - please correct); 

3. some editing error appears somewhere close to line 358 and also in line 367; 

4. the Authors use some fitting procedure to determine the best values for the constants C1 & C2 - please attach some RMS error and variance of this fitting - maybe some comparative worse approximators should be given to have a contrast for discussion. This seems to be one of the milestones in this research, so that should be discussed in detail. 
